# Computation and Sample Efficient Reinforcement Learning with Deep Bayesian Actor-Critic

## Abstract

Actor-critic methods in reinforcement learning leverage the action value function (critic) by temporal difference learning to be used as an objective for policy improvement for the sake of sample efficiency against on-policy methods. The well-known result, critic overestimation, is usually handled by pessimistic policy evaluation based on critic uncertainty, which may lead to critic underestimation. This means that pessimism is a sensitive parameter and requires careful tuning. Most methods employ an ensemble approach to represent the uncertainty of critic estimates, but it comes with the cost of computational burden. To mitigate the sample and computation inefficiency of the actor-critic approach, we propose a novel and simple algorithm in this paper, called Deep Bayesian Actor-Critic (DBAC), that employs Bayesian dropout and a heteroscedastic critic network instead of an ensemble to make the agent uncertainty-aware. To mitigate the overestimation bias of critic, pessimistic policy evaluation is conducted where pessimism is proportional to the uncertainty of predictions. Using dropout along with a distributional representation of the critic leads to more computation-efficient calculations. With empirically determined optimal pessimism and dropout regularization, only a single critic network is enough to achieve high sample and computation efficiency, with near SOTA performance.

## 1 Introduction

Reinforcement learning (RL) has witnessed significant progress with the emergence of deep neural networks (Mnih et al., 2013; 2015) in the last decade. However, sample efficiency is one of the main bottlenecks of widespread adaptation of RL into applications (Mendonca et al., 2019; Li et al., 2023a). On the other hand, efficiency is as important as sample efficiency (Chen et al., 2021a), especially to deploy RL agents into real-life applications such as robots (Zhao et al., 2020; Kormushev et al., 2013) and edge devices (Dai et al., 2022; Wei et al., 2022).

Actor-critic methods leverage off-policy samples to train critic, promising higher sample efficient learning than on-policy algorithms. Despite this advantage, they are usually stuck on poor performance due to a mismatch between behavioral and online policy since value estimates of actions, on which value network is never (or not enough) trained, are required. Finally, the erroneous value estimates are exploited and the algorithm suffers from critic overestimation that yields divergent (catastrophic) behavior (Thrun & Schwartz, 2014). This problem is also named as *deadly triad* ((Sutton & Barto, 2018; Van Hasselt et al., 2018)) indicating the instability emerges once function approximation, temporal difference, and off-policy learning are in the same method together.

Critic overestimation can also be explained by the limited generalization capability of the used function approximator (Korkmaz, 2024). Generalization is key to success for RL since the aim is to infer actions that are not similar to available data, i.e., it is expected to extrapolate a new policy to perform differently than a behavioral policy. If the critic of an actor-critic method would have strong *generalization capability*, it would estimate better values for out-of-distribution actions, and much less overestimation would occur.

## 1.1 Pessimism for Actor-Critic

One of the main solutions to the critic overestimation problem is to use a pessimistic objective. This way, it is acceptable to have poor generalization as long as the learner is aware of it. However, this requires to assess *epistemic uncertainty*, or in other words, the learner should be *uncertainty-aware*. Such models would identify more uncertainty for out-of-distribution actions and use lower bounds (conservative estimates) as objective at the cost of under-exploration. On the other hand, *optimism in the face of uncertainty* principle provides a reasonable exploration scheme in an on-policy setting as this encourages exploration of state-action space. However, for the off-policy actor-critic approach, it is only feasible to be optimistic on critic uncertainty for policy improvement, not for policy evaluation (Tasdighi et al., 2024a) since critic errors are exploited in a wrong and unrecoverable way in this phase.

In the literature, methods that assess their *epistemic uncertainty* (of transition model (Janner et al., 2019; Chua et al., 2018; Depeweg et al., 2016) or critic (Chen et al., 2021b; Hiraoka et al., 2021)) can obtain better performance and sample efficiency. Recently, for off-policy model-free actor-critic algorithms, epistemic uncertainty is estimated by using double network (Fujimoto et al., 2018; Haarnoja et al., 2018) or ensemble network (Chen et al., 2021b; Moskovitz et al., 2021). Hiraoka et al. (2021) found out that using Bayesian dropout (Srivastava et al., 2014; Gal & Ghahramani, 2016) contributes to epistemic uncertainty assessment but a small ensemble is still required. On the contrary, He et al. (2021) argued that a single critic network is enough when dropout is also used to evaluate Bellman backup.

Kuznetsov et al. (2020) claims *aleatoric uncertainty* is also responsible for overestimation since any randomness is exploited when the Bellman optimality operator ($\mathcal{T}^*$) is employed. For this purpose, they use ensemble networks for epistemic uncertainty and distributional representation (as quantiles) of network output for aleatoric uncertainty assessment.

Tasdighi et al. (2024a) and Moskovitz et al. (2021) showed that optimal pessimism (or optimism) is dependent on environment, task, and learning method. Once the *synergy* between learner and task emerges (good generalization), it does not need to be pessimistic for policy improvement and can trust its estimate. However, when most out-of-distribution estimates are not consistent with reality (poor generalization), they need to be pessimistic and should not trust their estimate unless sufficient observations are obtained. Therefore, *depending on the learner's generalization capability, it requires to balance pessimism effectively*.

## 1.2 Deep Bayesian Actor-Critic Algorithm

Actor-critic methods suffer from computation inefficiency when ensembles are employed for critic to track epistemic uncertainty. In addition, they also suffer from sample inefficiency when constant pessimism is employed. In this paper, we introduce a novel online actor-critic algorithm, named Deep Bayesian Actor-Critic (DBAC), specifically designed to address both sample and computation inefficiencies. DBAC is an off-policy maximum entropy actor-critic algorithm employing experience buffer to sample off-policy transition tuples, similar to SAC (Haarnoja et al., 2018).

Our first contribution is a theoretical analysis of critic overestimation under the maximum entropy actor-critic framework. As a result of this analysis, DBAC devises environment-specific pessimism hyper-parameter to be used upon *predictive uncertainty* of critic. Pessimistic updates are conducted in both policy evaluation and improvement phases to tackle critic overestimation. The optimal pessimism improves sample efficiency, yielding similar results to other methods that use a much higher update-to-data (UTD) ratio and ensembles.

As the second contribution, DBAC replaces critic ensemble with a single critic by employing *Bayesian dropout* (Srivastava et al., 2014; Gal & Ghahramani, 2016) on critic network to track epistemic uncertainty and distributional (heteroscedastic) value representation for predictive uncertainty. Although a heteroscedastic model is only able to capture aleatoric uncertainty (Kendall & Gal, 2017) in a supervised learning setting, it can also capture epistemic uncertainty caused by Bellman backup since the bootstrapped critic target has randomness sourced by dropout. Moreover, it also captures aleatoric uncertainty due to transition dynamics randomness and the non-stationary nature of the learning procedure. Lastly, heteroscedastic representations allow learning loss attenuation, making the critic loss more robust to noisy data (Kendall & Gal, 2017).

Using a single heteroscedastic critic network with dropout enhances computation efficiency compared to other methods that use ensembles.

The implementation is very simple and can be obtained by injecting dropout to networks, introducing a heteroscedastic critic network, and defining and pessimistic learning objective upon the well-known Soft Actor-Critic algorithm (Haarnoja et al., 2018). We conduct extensive experiments on standard RL benchmarks to evaluate the performance of DBAC compared to existing methods. Our results demonstrate the effectiveness of DBAC in achieving competitive performance to SOTA methods while requiring fewer computational resources and fewer samples, making it a promising approach for real-world RL applications.

## 2  Reinforcement Learning Preliminaries

### 2.1  Model-free Reinforcement Learning

In reinforcement learning language, the agent lives in a Markov Decision Process (MDP) which is represented by a tuple $\mathcal{M} = (\mathcal{S}, \mathcal{A}, d_0, \tau, R)$, where $\mathcal{S}$ is state space, $\mathcal{A}$ is action space, $d_0 \in \mathcal{P}(\mathcal{S})$ is initial state distribution, $\tau : \mathcal{S} \times \mathcal{A} \mapsto \mathcal{P}(\mathcal{S})$ is transition kernel and $R : \mathcal{S} \times \mathcal{A} \mapsto \mathbb{R}$ is reward function.

The initial state is sampled first, $s_0 \sim d_0(\cdot)$. At each time $t$ being on $s_t \in \mathcal{S}$; next state is sampled from the environment, $s_{t+1} \sim \tau(\cdot \mid s_t, a_t)$ depending on taken action $a_t \sim \pi(\cdot \mid s_t)$. Finally, reward is obtained, $r_t = R(s_t, a_t)$ by the reward function $R$. The ultimate goal of the agent is to derive a policy $\pi : \mathcal{S} \mapsto \mathcal{P}(\mathcal{A})$ to maximize discounted cumulative return, i.e., value function for a given state $s$,

$$V^\pi(s) = \mathbb{E}_{\pi,\tau}\Big[ \sum_{t=0}^\infty \gamma^t R(s_t, a_t) \Big| s_0 = s \Big]. \tag{1}$$

### 2.2  Maximum Entropy Actor-Critic

To promote random actions for exploration and algorithm robustness, maximum entropy framework introduces policy entropy bonus into value functions (Haarnoja et al., 2017; 2018),

$$V^\pi(s) = \mathbb{E}_{\pi,\tau}\Big[ \sum_{t=0}^\infty \gamma^t R(s_t, a_t) - \alpha \log \pi(a_t|s_t) \Big| s_0 = s \Big], \tag{2}$$

$$Q^\pi(s,a) = R(s,a) + \mathbb{E}_{\pi,\tau}\Big[ \sum_{t=1}^\infty \gamma^t R(s_t, a_t) - \alpha \log \pi(a_t|s_t) \Big| s_0 = s, a_0 = a \Big]. \tag{3}$$

Learning iterates between solving policy evaluation and policy improvement. For the definition of critic, Bellman backup operator $\mathcal{T}^\pi$ is defined,

$$\mathcal{T}^\pi Q(s,a) = R(s,a) + \gamma \mathbb{E}_{\substack{s' \sim \tau(\cdot|s,a) \\ a' \sim \pi(\cdot|s')}} \big[ Q(s', a') - \alpha \log \pi(a' \mid s') \big], \tag{4}$$

and the critic is expected to remain same if this operator applied on itself, i.e., $Q^\pi(s,a) = \mathcal{T}^\pi Q^\pi(s,a)$. The policy evaluation phase minimizes the temporal difference, i.e., the difference between $Q$ and $\mathcal{T}^\pi Q(s,a)$ to satisfy this condition. Therefore, the Bellman backup $\mathcal{T}^\pi Q(s,a)$ is also called the temporal difference (TD) target.

The optimal policy is defined as softmax over optimal critic,

$$\pi^*(\cdot \mid s) = \arg\min_\pi \text{KL}\Big( \pi(\cdot \mid s) \Big\| \frac{\exp(\alpha^{-1} Q^*(s, \cdot))}{\int_\mathcal{A} \exp(\alpha^{-1} Q^*(s, a)) da} \Big). \tag{5}$$

The policy improvement phase solves Equation 5 for available critic $Q$ instead of $Q^*$. After sufficient iteration, both policy and critic converge to optimality in the ideal case. The optimal critic must satisfy Bellman optimality, $Q^*(s,a) = \mathcal{T}^* Q^*(s,a)$, where Bellman optimality operator $\mathcal{T}^*$ turns out to have following form (Equation 5 from Haarnoja et al. (2017)),

$$\mathcal{T}^* Q(s,a) = R(s,a) + \gamma \mathbb{E}_{s' \sim \tau(\cdot|s,a)} \Big[ \alpha \log \Big( \int_\mathcal{A} \exp(\alpha^{-1} Q(s', a')) da' \Big) \Big]. \tag{6}$$

# 3 Modeling Aleatoric and Epistemic Uncertainties

In this part, we explain the differences between two main types of uncertainties, *aleatoric* and *epistemic* uncertainty (Der Kiureghian & Ditlevsen, 2009; Kendall & Gal, 2017; Gal et al., 2016b). Most deep learning methods model either epistemic or aleatoric uncertainty alone (Gal et al., 2016b), whereas modeling both has fundamental importance for reliable and robust predictions.

## 3.1 Aleatoric Uncertainty

This type of uncertainty comes from the inherent randomness within the data itself. It is sometimes called statistical or data uncertainty. Even if more data are collected, aleatoric uncertainty is unavoidable and cannot be reduced, because it is an intrinsic part of the process being modeled. For instance, this could be due to measurement errors or natural variability in the data. Additionally, uncertainty due to lack of learning capacity may also appear as aleatoric uncertainty from the model's side, as it cannot be reduced by collecting more data. In other words, the agent cannot decide true deterministic output just because it is not capable of doing it and assigns a non-deterministic distribution as output.

For regression problems in deep learning setting, we can model this by having a heteroscedastic network (with parameter $\theta$) that outputs a normal distribution $\mathcal{N}(\mu_\theta(x), \sigma_\theta^2(x))$, where both mean and variance depends on input $x$ (Lakshminarayanan et al., 2017; Kendall & Gal, 2017). Given a dataset $\mathcal{D} = \{(x_i, y_i)\}_{i=1}^N$, the loss function for training the network can be derived from the negative log-likelihood of the normal distribution,

$$\mathcal{L}_\theta(\mathcal{D}) = -\log p(\mathcal{D} \mid \theta) = \frac{1}{N} \sum_{i=1}^N \left( \frac{1}{2\sigma_\theta^2(x_i)} (y - \mu_\theta(x_i))^2 + \frac{1}{2} \log \sigma_\theta^2(x_i) \right) + \frac{1}{2} \log 2\pi. \tag{7}$$

## 3.2 Epistemic Uncertainty

This type of uncertainty arises from a lack of model knowledge. Also known as model uncertainty, epistemic uncertainty can be reduced by gathering more data, refining the model, or simply using better modeling techniques. It reflects the uncertainty in the model's parameters and structure due to insufficient training data, or incomplete understanding of the underlying process.

In deep learning context, Bayesian neural networks (BNNs) provide a way to model epistemic uncertainty. In BNNs, there is a prior distribution $p(\theta)$ over the network parameters $\theta$. Given the training data $\mathcal{D}$, we can compute the posterior distribution over the parameters,

$$p(\theta \mid \mathcal{D}) = \frac{p(\mathcal{D} \mid \theta)p(\theta)}{p(\mathcal{D})}. \tag{8}$$

In deep variational inference, we approximate posterior $p(\theta \mid \mathcal{D})$ by a neural network $q_w(\theta)$ parameterized by $w$. The objective is to maximize the posterior distribution, which is same as minimizing the evidence lower bound (ELBO) as loss function,

$$\mathcal{L}_w(\mathcal{D}) = \mathbb{E}_{q_w(\theta)} \left[ -\log p(\mathcal{D} \mid \theta) \right] + \mathrm{KL}(q_w(\theta) \| p(\theta)), \tag{9}$$

combining likelihood of data where parameter is sampled by posterior $q_w(\theta)$, and Kullback-Leibler (KL) divergence from posterior $q_w(\theta)$ to prior $p(\theta)$.

During prediction, we marginalize over the posterior distribution of the parameters to obtain the predictive distribution $p(y \mid x, \mathcal{D})$, and it is often approximated by sampling several sets of parameters from the posterior distribution $p(\theta \mid \mathcal{D})$ representing epistemic uncertainty, and averaging the predictions from distribution $p(y \mid x, \theta)$ representing aleatoric uncertainty,

$$p(y \mid x, \mathcal{D}) = \int p(y \mid x, \theta)p(\theta \mid \mathcal{D})d\theta. \tag{10}$$

**Practical Implementation with Monte Carlo Dropout**   Monte Carlo dropout is a practical method to approximate Bayesian inference in neural networks (Gal & Ghahramani, 2016; Gal et al., 2017). During training, dropout is applied and the model learns to make predictions with dropout active. The loss function in this setting typically remains the same as the standard loss (e.g., negative log-likelihood) but with dropout applied. Multiple stochastic forward passes with dropout enabled are performed during inference to approximate the predictive distribution, capturing epistemic uncertainty.

### 3.3 Uncertainty in Reinforcement Learning

In reinforcement learning, uncertainty representation for the value of a policy carries fundamental importance, especially in the presence of approximation (Bellemare et al., 2017). This can be conducted by atoms (Bellemare et al., 2017), quantiles (Dabney et al., 2018), and a probability distribution (Tang et al., 2019; Yang et al., 2021). On the other hand, the representation of critic output as a distribution only allows us to assess aleatoric uncertainty. To have a reasonable predictive uncertainty, epistemic uncertainty should also be estimated. For this, ensembles (Chen et al., 2021b; Kuznetsov et al., 2020), Bayesian neural networks (Tasdighi et al., 2024b) or Bayesian dropout (Hiraoka et al., 2021) can be used.

The predictive value uncertainty representation is the key to adjusting *optimism* vs *pessimism* balance, i.e., *risk seeking* vs *risk averse* behavior. Especially, it is important to balance critic overestimation-underestimation, as the main purpose of this work and similar studies.

## 4   Quantifying Overestimation for Sub-Gaussian Critic Distributions

In this part, we analyze how estimation error causes overestimation due to policy improvement. Assuming the policy improvement step is successful and given $Q$, the target used to update the critic in maximum entropy framework is simply Bellman backup $\mathcal{T}^\pi Q$, i.e., Bellman backup operator applied on $Q$. The definition uses a deterministic function $Q$ (Equation 4) while the critic is only known with some uncertainty, not exactly, represented as a distribution over returns $\mathcal{Q}$. Therefore, we define stochastic Bellman backup $\mathcal{T}^\pi \mathcal{Q}$ as follows;

$$\mathcal{T}^\pi \mathcal{Q}(s,a) = R(s,a) + \gamma \mathbb{E}_{\substack{q \sim \mathcal{Q} \\ s' \sim \tau(\cdot|s,a) \\ a' \sim \pi(\cdot|s')}} \Big[ q(s',a') - \alpha \log \pi(a' \mid s') \Big]. \tag{11}$$

Similarly, we define stochastic Bellman update $\mathcal{T}^* \mathcal{Q}$, i.e., Bellman optimality operator applied on $\mathcal{Q}$ as follows;

$$\mathcal{T}^* \mathcal{Q}(s,a) = R(s,a) + \gamma \mathbb{E}_{\substack{q \sim \mathcal{Q} \\ s' \sim \tau(\cdot|s,a)}} \Big[ \alpha \log \Big( \int_{\mathcal{A}} \exp(\alpha^{-1} q(s',a')) da' \Big) \Big]. \tag{12}$$

Now, we analyze overestimation bias, similar to the work of Chen et al. (2021b) and Lan et al. (2020) but in the soft learning framework instead of discrete actions. Our main purpose is to find the source of critic overestimation and to devise a pessimistic Bellman operator to prevent overestimation.

**Definition 4.1.** *A random variable $X \in \mathbb{R}$ with mean $\mu = \mathbb{E}[X]$ is called sub-Gaussian with variance proxy $\sigma^2$ if its moment generating function satisfies*

$$\mathbb{E}[\exp(\lambda X)] \leq \exp\big(\lambda \mu + \frac{1}{2}\lambda^2 \sigma^2\big), \quad \forall \lambda \in \mathbb{R}. \tag{13}$$

Let $\mu(s,a) = \mathbb{E}_{q \sim \mathcal{Q}}[q(s,a)]$. We define overestimation error as difference between $\mathcal{T}^* \mathcal{Q}$ and average $\mathcal{T}^* \mu$ as $\epsilon$,

$$\epsilon(s,a) = \mathcal{T}^* \mathcal{Q}(s,a) - \mathcal{T}^* \mu(s,a). \tag{14}$$

Ideally, $\epsilon(s,a)$ should be zero if there is no overestimation, which is not the case due to uncertainty. To quantify it, we assume that critic distribution $\mathcal{Q}(s,a)$ is sub-Gaussian with variance proxy $\sigma^2(s,a)$, representing uncertainty. Finally, we possess Theorem 4.1.

**Theorem 4.1** (Overestimation quantification for sub-Gaussian critics)**.** *Let estimated critic distribution* $\mathcal{Q} \in \mathcal{P}(\Omega_Q)$ *be sub-Gaussian with mean* $\mu$ *and variance proxy* $\sigma^2$*. Then,*

$$\mathcal{T}^*\mathcal{Q}(s,a) \leq R(s,a) + \gamma\mathbb{E}_{s'\sim\tau(\cdot|s,a)}\Big[\alpha\log\Big(\int_{\mathcal{A}}\exp(\alpha^{-1}\mu(s',a') + \frac{1}{2}\alpha^{-2}\sigma^2(s',a'))da'\Big)\Big]. \qquad (15)$$

*In addition, overestimation due to uncertainty of estimated distribution* $\mathcal{Q}$*, denoted as* $\epsilon$*, is upper bounded for Bellman updates,*

$$\epsilon(s,a) \leq \frac{\gamma}{2\alpha}\mathbb{E}_{s'\sim\tau(\cdot|s,a)}\Big[\max_{a'}\sigma^2(s',a')\Big]. \qquad (16)$$

**Corollary 4.1.1** (Pessimistic critic target)**.** *Given estimated critic distribution* $\mathcal{Q}$*, using shifted distribution* $\tilde{\mathcal{Q}} = \mathcal{N}(\tilde{\mu}, \tilde{\sigma}^2)$ *for Bellman updates, where mean is shifted* $\tilde{\mu} = \mu - \beta\sigma$ *with same variance proxy* $\tilde{\sigma}^2 = \sigma^2$*, prevents overestimation as long as* $\beta \geq \max_{(s',a')}\frac{1}{2}\alpha^{-1}\sigma(s',a')$*.*

The source of overestimation and necessity of pessimistic training is revealed in Theorem 4.1 and Corollary 4.1.1. Although using pessimistic critic targets for training critic and policy is not a new idea (Moskovitz et al., 2021; Kuznetsov et al., 2020; Chen et al., 2021b), the question of how to determine pessimism ($\beta$) remains. For this, Moskovitz et al. (2021) had shown that optimal pessimism/optimism depends on the environment, and stated that *estimation bias depends on overall context in which a learner is embedded.* Similarly, we argue that *estimation bias depends on the generalization capability of the learner.*

## 5 Deep Bayesian Actor-Critic

In this section, we discuss key mechanisms needed for computation and sample efficient actor-critic learning and propose our algorithm *Deep Bayesian Actor-Critic*. This algorithm employs *single* critic network which captures uncertainty instead of ensembling. For this, it employs Bayesian dropout within the critic network and learns probability distribution as output representing *predictive uncertainty* to be used to evaluate the pessimistic Bellman backup. In addition, the policy network has also dropout within layers to regularize its improvement phase. Neural architectures of critic and policy are illustrated in Appendix D.

### 5.1 Heteroscedastic Critic

Heteroscedastic networks output probability distribution instead of a point estimate and are originally designed to model aleatoric uncertainty of underlying phenomena (Kendall & Gal, 2017; Lakshminarayanan et al., 2017). In addition to this property, modeling output as a distribution allows the network to learn loss attenuation and makes learning robust to noisy data (Kendall & Gal, 2017). In our setting, the objective is to fit a distribution of Bellman backup which includes epistemic randomness of the bootstrap estimate, and aleatoric randomness of state transition and non-stationary learning procedure. Therefore, most of the critic uncertainty is modeled this way, and this can be used for pessimistic updates of critic and policy.

### 5.2 Pessimistic Objective

Like most algorithms, the natural way to inhibit overestimation is by employing pessimistic critic updates. Assuming critic value distribution is normal (still sub-Gaussian), we can use modified pessimistic distribution $\tilde{\mathcal{Q}} = \mathcal{N}(\mu - \beta\sigma, \sigma^2)$ from Corollary 4.1.1, but it would be overpessimistic for higher $\beta$ values which are required to guarantee overestimation prevention. However, there are many other factors affecting the optimal pessimism level. For example, policy improvement is slower than policy evaluation phase in actor-critic methods, decreasing possible overestimation. In addition, the real variance might be lower than the estimated variance since the estimation is not calibrated. Lastly, overestimation may not need to be fully prevented and slight optimistic updates may promote exploration. For this purpose, we state that $\beta$ simply stands as a pessimism parameter to be tuned for each environment and learning hyper-parameters and it can be small depending on the learning process. At the end, we define the pessimistic Bellman update $\tilde{\mathcal{T}}^*\mathcal{Q}(s,a)$ as follows;

$$\tilde{\mathcal{T}}^* \mathcal{Q}(s,a) = \mathcal{T}^* \tilde{\mathcal{Q}}(s,a) = R(s,a) + \gamma \mathbb{E}_{\substack{q \sim \tilde{\mathcal{Q}} \\ s' \sim \tau(\cdot|s,a)}} \left[ \alpha \log \left( \int_{\mathcal{A}} \exp(\alpha^{-1} q(s',a')) da' \right) \right]. \tag{17}$$

### 5.3 Dropout Regularization

Dropout regularization (Srivastava et al., 2014) allows to capture the probabilistic nature of a network, representing Bayesian neural networks (Gal & Ghahramani, 2016). It is also equivalent to representing the model as an ensemble since each sampled weight set of the network corresponds to a sub-model (He et al., 2021). For this purpose, DBAC employs dropout regularization for both critic and policy networks.

For the learning phase, it prevents policy and critic from overfitting and improves generalization. More importantly, temporal difference (TD) targets are evaluated with dropout which has randomness of epistemic uncertainty, leading up to be learned in heteroscedastic critic distribution. In the end, overall predictive uncertainty can be quantified as a simple normal distribution and used to construct a pessimistic objective.

### 5.4 Layer Normalization

Layer Normalization (Ba et al., 2016) is a normalization method applied to feature dimensions of activations. It has a regularization effect and prevents possible numerical instabilities in training time. In DBAC, we implement Layer Normalization after all hidden activations of critic and policy networks, similar to Hiraoka et al. (2021).

### 5.5 Algorithm Summary

Finally, we present the Deep Bayesian Actor-Critic (DBAC) algorithm using the results of analyses from previous sections. Unlike previous methods, we parameterize critic $\mathcal{Q}_\theta$ as a single network by parameter set $\theta$ and policy $\pi_\phi$ as another single network by parameter set $\phi$, where both networks have probability distribution as outputs, i.e., networks represent distributions over values and actions. Policy outputs a `tanh` transformed normal distribution to bound actions to $[-1,1]$. Networks illustrations are available in Appendix D. Note that the bar notation stands for the lagged network with non-trained parameters. DBAC is summarized in Algorithm 1 with gradient descent but Adam optimizer (Kingma & Ba, 2014) is used in our experiments.

**Critic learning** Critic network predicts cumulative return with some uncertainty. Using transition tuples from experience replay as batch, $\mathcal{D}_b = \{(s_i, a_i, r_i, s_i', \texttt{done}_i)\}_{i=1}^{N_b}$, temporal difference (TD) target $q_i^{TD}$, representing Bellman backup, is $\beta$-pessimistic,

$$q_i^{TD} = r_i + \gamma(\mu_{\bar{\theta}}(s_i', \tilde{a}_i') - \beta \sigma_{\bar{\theta}}(s_i', \tilde{a}_i') - \alpha \log \pi_\phi(s_i', \tilde{a}_i'))(\neg \texttt{done}_i), \quad \tilde{a}_i' \sim \pi_\phi(\cdot \mid s_i'). \tag{18}$$

Learning objective is cross-entropy loss (log loss),

$$\mathcal{L}_\theta(\mathcal{D}_b) = \frac{1}{N_b} \sum_{i=1}^{N_b} -\log \mathcal{Q}_\theta(q_i^{TD} \mid s_i, a_i). \tag{19}$$

Theoretically, critic distribution is not restricted to any type but sub-Gaussian. For simplicity, we modeled the critic to be represented as a normal distribution, i.e., $\mathcal{Q}_\theta = \mathcal{N}(\mu_\theta, \sigma_\theta^2)$ in this work. In this case, the cross entropy loss becomes as follows;

$$\mathcal{L}_\theta(\mathcal{D}_b) = \frac{1}{2} \log 2\pi + \frac{1}{N_b} \sum_{i=1}^{N_b} \left( \frac{1}{2} \log \sigma_\theta^2(s_i, a_i) + \frac{(q_i^{TD} - \mu_\theta(s_i, a_i))^2}{2\sigma_\theta^2(s_i, a_i)} \right). \tag{20}$$

In this loss, $q_i^{TD}$ is simply a bootstrapped estimate, used in temporal difference methods. The major difference is that we learn critic with cross entropy (log) loss and TD target $q_i^{TD}$ has also epistemic randomness sourced by dropout.

**Lagged critic for TD target**  When the trained critic network is also used in calculating the target value, the critic training is prone to divergence (Li et al., 2023b). For this, a common approach is to use another critic network to evaluate TD target (Mnih et al., 2013). Similar to Lillicrap et al. (2015), Fujimoto et al. (2018), and Haarnoja et al. (2018), we use a lagged critic network with non-trainable parameters for TD target evaluations as demonstrated in Equation 18. The network parameters are only updated by Polyak averaging of main critic network weights through learning steps. This strategy is important to ensure the stability of temporal difference learning.

**Policy learning**  The policy improvement objective has a very similar form to SAC algorithm (Haarnoja et al., 2018) except using standard deviation to construct $\beta$-pessimistic objective instead of the minimum of double critic predictions. Using only states from experience replay as batches $\mathcal{D}_b = \{(s_i)\}_{i=1}^{N_b}$ with batch size $N_B$, loss function for policy network is as follows;

$$\mathcal{L}_\phi(\mathcal{D}_b) = \frac{1}{N_b} \sum_{i=1}^{N_b} \mathbb{E}_{a \sim \pi_\phi(\cdot | s_i)} \big[ \mu_\theta(s_i, a) - \beta \sigma_\theta(s_i, a) - \alpha \log \pi_\phi(a \mid s_i) \big]. \tag{21}$$

**Automatic temperature tuning**  Inspired from Haarnoja et al. (2018), we also employed automatic temperature tuning. Using constant temperature results in different policies if the reward magnitude changes. To mitigate this, Haarnoja et al. (2018) proposed a policy entropy constraint, representing temperature as the Lagrange multiplier of the constraint. Given target entropy $\bar{\mathcal{H}}$ as hyper-parameter, the loss function related to this constraint is as follows;

$$\mathcal{L}_\alpha(\mathcal{D}_b) = -\alpha\bar{\mathcal{H}} + \alpha \sum_{i=1}^{N_b} \mathbb{E}_{a \sim \pi_\phi(\cdot | s_i)} \big[ -\log \pi_\phi(a \mid s_i) \big]. \tag{22}$$

---

**Algorithm 1** Deep Bayesian Actor-Critic

---

**Require:** Environment `env`
**Require:** Experience buffer $\mathcal{D}$
**Require:** Critic $\mathcal{Q}_\theta$, lagged critic $\mathcal{Q}_{\bar{\theta}}$, policy $\pi_\phi$, all with dropout
**Require:** Initial temperature $\alpha$, target entropy $\bar{\mathcal{H}}$
**Require:** Pessimism $\beta$
**Require:** Learning rates $\eta_Q, \eta_\pi, \eta_\alpha$, Polyak parameter $\rho$
**Require:** Total training steps $N$, batch size $N_b$

  $s \sim$ `env.reset()`                                           ▷ Reset the environment
  **for** $N$ timesteps **do**
    $a \sim \pi_\phi(\cdot \mid s)$                                     ▷ Sample action
    $r, s', $ `done` $\sim$ `env.step`$(a)$                      ▷ Act on environment
    $\mathcal{D} \leftarrow \mathcal{D} \cup (s, a, r, s', $ `done`$)$             ▷ Record transition tuple
    **if** `done` **then** $s \leftarrow s'$ **else** $s \sim$ `env.reset()`    ▷ State transition or reset
    **for** $G$ gradient steps **do**
      $\mathcal{D}_b = \{(s_i, a_i, r_i, s_i', $ `done`$_i)\}_{i=1}^{N_b} \sim \mathcal{D}$    ▷ Sample minibatch for training
      $\tilde{a}_i' \sim \pi_\phi(\cdot \mid s_i'), \quad \forall i \in \{1, 2, ..., N_b\}$    ▷ Sample next actions
      $q_i^{TD} = r_i + \gamma(\mu_{\bar{\theta}}(s_i', \tilde{a}_i') - \beta\sigma_{\bar{\theta}}(s_i', \tilde{a}_i'))(\neg$`done`$_i), \quad \forall i \in \{1, 2, ..., N_b\}$   ▷ Build TD targets
      $\theta \leftarrow \theta - \eta_Q \nabla_\theta \Big( \frac{1}{N_b} \sum_{i=1}^{N_b} -\log \mathcal{Q}_\theta(q_i^{TD} \mid s_i, a_i) \Big)$   ▷ Update critic
      $\phi \leftarrow \phi - \eta_\pi \nabla_\phi \Big( \frac{1}{N_b} \sum_{i=1}^{N_b} \mathbb{E}_{a \sim \pi_\phi(\cdot|s_i)} \big[ \mu_\theta(s_i, a) - \beta\sigma_\theta(s_i, a) - \alpha\log\pi_\phi(a \mid s_i) \big] \Big)$   ▷ Update policy
      $\alpha \leftarrow \alpha - \eta_\alpha \nabla_\alpha \Big( -\alpha\bar{\mathcal{H}} + \alpha \sum_{i=1}^{N_b} \mathbb{E}_{a \sim \pi_\phi(\cdot|s_i)} \big[ -\log\pi_\phi(a \mid s_i) \big] \Big)$   ▷ Update temperature
      $\bar{\theta} \leftarrow \rho\bar{\theta} + (1 - \rho)\theta$                   ▷ Update target critic network
    **end for**
  **end for**

---

# 6 Experiments

Our experiments aim to investigate whether enhancing off-policy actor-critic methodology with DBAC can improve their sample and computation efficiency on difficult continuous-control benchmarks. For this purpose, DBAC is compared to similar competitive algorithms; TQC (Kuznetsov et al., 2020), DROQ (Hiraoka et al., 2021), SAC (Haarnoja et al., 2018) and TOPSAC, which is SAC variant of TOP algorithm (Moskovitz et al., 2021), where only exploration scheme is changed to maximum entropy policy. All algorithm results are obtained using in-house code with the same network architectures (including layer normalization) to make a fair comparison. We included DROQ algorithm with UTD ratio ($G$) equal to 1 and 5, although it is equal to 20 in the original paper. Additionally, two ablations studies are conducted to examine the effectiveness of different levels of pessimism and dropout rates may vary across different environments. Lastly, the effect of Layer Normalization is not surveyed since it is done by Hiraoka et al. (2021) extensively for DROQ algorithm.

Through Gymnasium API (Towers et al., 2023), six well-known MuJoCo environments (Todorov et al., 2012) are used for comparison as they are tested by most algorithms in the literature. Hyper-parameters per environment can be found in Table 4 of Appendix C. For all experiments, PyTorch (version 2.2.2) (Paszke et al., 2019) is used. Please refer to Appendix E for codebase.

**Evaluation protocol**   After each 1000 time steps, we execute a single test episode using the online policy and measure its performance by calculating the total reward accumulated during the episode. For each environment, the number of training steps is different to keep the training duration short. Total training steps are taken from REDQ (Chen et al., 2021b) and MBPO (Janner et al., 2019), except `InvertedDoublePendulum-v4` since it is not available on those papers.

**Learning curves**   Specified environments are trained through a fixed number of environment interactions, repeated 5 times to assess the stability of the algorithm shown by mean and standard deviation. Further experimental details are presented in Appendix C. Mean and standard deviations on the last evaluation episode are summarized in Table 1. Additionally, average returns through all learning processes averaged over random seeds are summarized in Table 2. In Figure 1, the performance of DBAC is shown against previously mentioned SOTA algorithms for 6 tasks, where important hyper-parameters used for DBAC and TQC are summarized in Table 5. Additionally, value estimation errors are presented in Figure 2. The bold lines represent the average, while the shaded area indicates the standard deviation (to represent randomness) of the total reward across evaluation episodes.

Table 1: Episodic return over five training runs on MuJoCo tasks at the end of training. $\pm$ sign denotes one standard deviation across trials. The first and second best methods are highlighted in blue and red.

| Env | # steps | DBAC | DROQ G=1 | DROQ G=5 | SAC | TOPSAC | TQC |
|---|---|---|---|---|---|---|---|
| `InvertedDoublePendulum-v4` | 50k | $9354 \pm 2$ | $8036 \pm 2634$ | $9349 \pm 6$ | $7899 \pm 2884$ | $7613 \pm 3484$ | $9354 \pm 2$ |
| `Walker2d-v4` | 300k | $4901 \pm 87$ | $3896 \pm 231$ | $831 \pm 298$ | $490 \pm 280$ | $2739 \pm 1310$ | $4368 \pm 296$ |
| `Hopper-v4` | 125k | $2886 \pm 376$ | $1550 \pm 1006$ | $1612 \pm 1154$ | $540 \pm 137$ | $1752 \pm 1145$ | $2789 \pm 860$ |
| `Humanoid-v4` | 300k | $4454 \pm 1672$ | $1196 \pm 789$ | $980 \pm 427$ | $1482 \pm 799$ | $1150 \pm 639$ | $3303 \pm 2330$ |
| `HalfCheetah-v4` | 400k | $8897 \pm 399$ | $6925 \pm 1208$ | $7632 \pm 995$ | $7303 \pm 742$ | $8049 \pm 1300$ | $9244 \pm 611$ |
| `Ant-v4` | 300k | $5298 \pm 281$ | $876 \pm 946$ | $4068 \pm 1427$ | $2638 \pm 1315$ | $1700 \pm 1527$ | $5926 \pm 163$ |

Table 2: Average episodic return through learning procedure and over five training runs on MuJoCo tasks. The first and second best methods are highlighted in blue and red.

| Env | # steps | DBAC | DROQ G=1 | DROQ G=5 | SAC | TOPSAC | TQC |
|---|---|---|---|---|---|---|---|
| `InvertedDoublePendulum-v4` | 50k | 6235.99 | 5967.47 | 6812.88 | 5578.03 | 5750.87 | 5624.03 |
| `Walker2d-v4` | 300k | 3166.3 | 1490.9 | 1300.85 | 1371.76 | 1822.86 | 2963.89 |
| `Hopper-v4` | 125k | 1122.34 | 502.25 | 1024.69 | 421.8 | 765.93 | 1193.79 |
| `Humanoid-v4` | 300k | 2271.2 | 952.87 | 994.22 | 1182.69 | 833.41 | 1880.35 |
| `HalfCheetah-v4` | 400k | 6866.44 | 5675.06 | 6483.58 | 6140.46 | 6149.67 | 7149.85 |
| `Ant-v4` | 300k | 2742.98 | 983.09 | 2046.47 | 1582.81 | 1140.96 | 3056.95 |

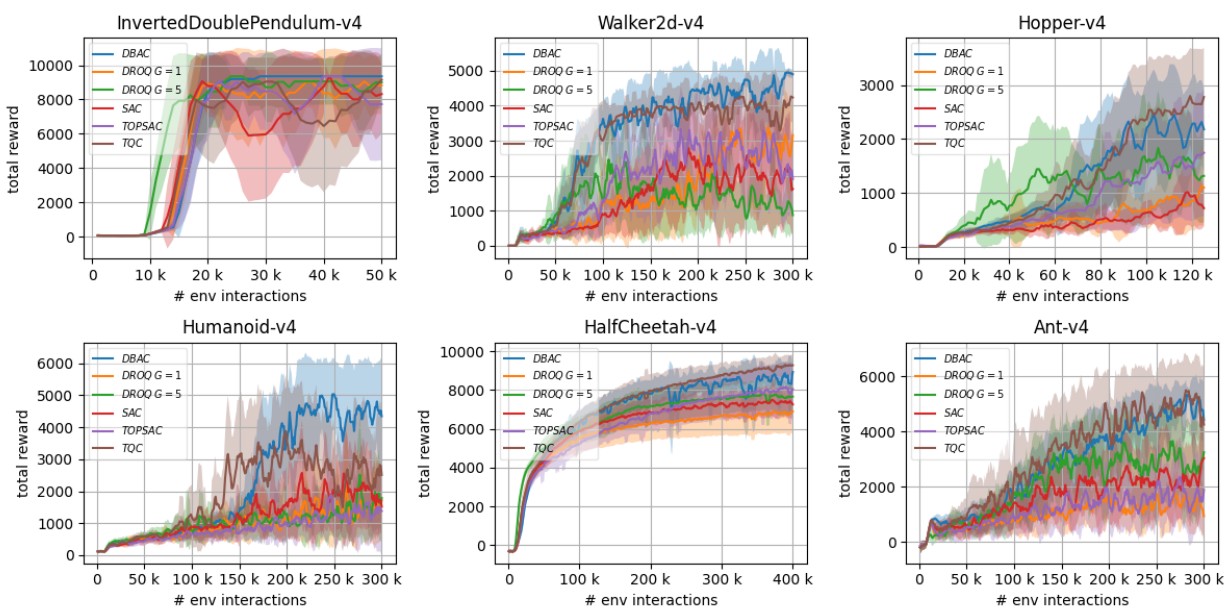

Figure 1: Main learning curves of DBAC and other algorithms. The standard deviation is represented by the shaded areas, while the average return across evaluation episodes is shown by solid curves. See specific hyper-parameters from Table 5.

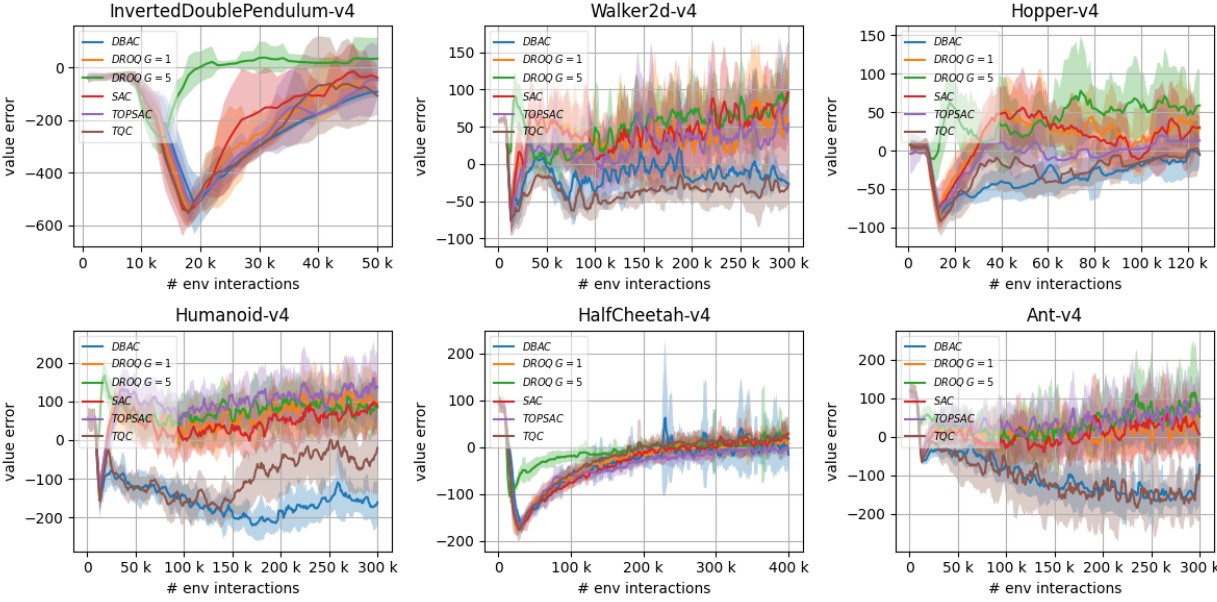

Figure 2: Estimation error of DBAC and other algorithms on the beginning of episodes. The standard deviation is represented by the shaded areas, while the average errors across evaluation episodes are shown by solid curves. See specific hyper-parameters from Table 5.

**Sample efficiency**   As seen from Figure 1, DBAC outperforms all other algorithms except TQC for `HalfCheetah-v4`, `Ant-v4` and `Hopper-v4` in terms of sample efficiency. The main explanation is in Figure 2, as other algorithms except TQC suffer from positive overestimation bias, where DBAC handles it by using a pessimism level specifically selected for each environment. It is also the same for TQC algorithm, as we found the number of quantiles to drop per network by trial-and-error to represent pessimism. Also as seen in Table 2, DBAC also performs well not only at the end of training but also during whole learning time along with TQC.

**Computation efficiency**   As wall-clock time statistics vary depending on computing units, we present the number of critic networks and number of critic backpropagations per time step in Table 3. Note that policy is a single and same network through all methods, and only the input and output layers of the critic are different which has an insignificant effect on the number of training parameters. Although TQC performs slightly better than DBAC on sample efficiency for some environments, DBAC uses fewer parameters and consumes fewer computation resources compared to TQC as DBAC uses only a single critic network with a UTD ratio of 1. Therefore, DBAC outperforms other algorithms in computation efficiency.

Table 3: Number of critic networks and backprops per time step. Each critic has almost same size.

|                   | DBAC | DROQ G=1 | DROQ G=5 | SAC | TOPSAC | TQC |
|-------------------|------|----------|----------|-----|--------|-----|
| # critic network  | 1    | 2        | 2        | 2   | 2      | 5   |
| # critic backprop | 1    | 2        | 10       | 2   | 2      | 5   |

However, the key behind DBAC's performance depends on two main hyper-parameters, pessimism and dropout rate of critic and policy networks. Therefore, we conduct sensitivity analysis for this purpose in the following sections.

## 6.1   Ablation Study 1: Pessimism Sensitivity

To investigate sensitivity of DBAC to pessimism parameter $\beta$, we run DBAC on all environment by varying $\beta$. Results are available in Appendix B.1. As shown in Figure 3, results indicate that the optimal $\beta$ parameter is neither small nor big number and it is a sensitive parameter. In Figure 4, higher $\beta$ yields lower error, consistent with our assumptions. Therefore, it should be determined carefully to guarantee better performance. As we stated earlier, excess pessimism paves the way to underestimation, whereas lack of it causes critic overestimation which is inherent to actor-critic methods.

In addition, varied pessimism levels do not affect results significantly in `HalfCheetah-v4` and `InvertedDoublePendulum-v4` environments compared to others and have similar value error curves. In other environments, score curves are worse if value estimations tend to be positive, meaning that critic overestimation is not handled well. This is observed in less pessimistic settings. On the other hand, score curves are again worse when error curves are negative and far from zero, meaning that the learner is stuck on critic underestimation caused by high pessimism. As an extra observation, for `Humanoid-v4`, sensitivity is much higher, and episodic value estimate diverged for $\beta = 1.1$ and $\beta = 1.2$, as shown in Figure 4.

In the end, pessimism sensitivity varies for different environments, possibly because of varying task difficulties. For easier tasks, less pessimism is enough but difficult tasks require significant pessimism. This parameter stands as the major bottleneck of DBAC and can only be determined by this heuristic for now.

## 6.2   Ablation Study 2: Dropout Sensitivity

In addition to pessimism, dropout rate is also an important parameter as it determines epistemic uncertainty. Results are available in Appendix B.2. As it can be seen from Figure 5, for most environments, dropout rate 0.01 is best, except `HalfCheetah-v4` and `Walker2d-v4`. This is the reason behind using zero dropout for the mentioned environments in the main comparison study (Table 5). This hyper-parameter is also sensitive and requires careful tuning. Although optimal dropout rate may also depend on the neural network architecture and environment, it is known that mentioned environments are relatively easier compared to

others in terms of overestimation issues, which is obvious from high performance with relatively primitive algorithms like Vanilla SAC (Haarnoja et al., 2018), TD3 (Fujimoto et al., 2018) and DDPG (Lillicrap et al., 2015). Therefore, we believe that dropout for those environments only slows down learning as they do not suffer from the critic overestimation problem sourced by epistemic uncertainty too much.

Moreover, zero dropout yields to relatively positive episodic value error as demonstrated in Figure 6. For `HalfCheetah-v4` and `Ant-v4`, both error curves and episodic returns have similar behaviour, while corresponding curves for `Hopper-v4` and `Humanoid-v4` differ together. Although pessimism is kept constant through dropout changes, dropout affects episodic value estimates since predictive uncertainty is also affected by the randomness of the temporal difference target sourced by dropout. However, this is probably not the single reason since dropout alters the overall learning procedure.

Another interesting result is that for all zero dropout experiments, DBAC is still on par with SAC, DROQ, and TQC (for some environments) in terms of sample efficiency, meaning that only representing the critic as a distribution, and learning by hand-tuned pessimism level removes the necessity to use double or ensemble critic network.

### 6.3 Ablation Study 3: Experiments on Stochastic Environments

As DBAC also captures aleatoric uncertainty of critic, it is also tested along with TQC on stochastic environments with continuous action space, `BipedalWalker-v3` and `BipedalWalkerHardcore-v3` from Gymnasium library (Towers et al., 2023). These environments have stochastic transition dynamics since the terrain where the walker walks is randomly generated. Results are available in Appendix B.3. $\beta$ parameter is swept for three values while $n_{drop}$ parameter of TQC is determined by trial-and-error to get best performance. Learning is terminated after 500k although the task is not completed but both algorithms tend to learn the task as seen from Figure 7 and the episodic value error curve is summarized in Figure 8.

Best performance is obtained when $\beta = 0.1$ and $\beta = 0.5$ for `BipedalWalkerHardcore-v3` and `BipedalWalker-v3` respectively, indicating that different pessimism levels are required to solve even similar environments. TQC demonstrates slightly better performance in both environments with $n_{drop} = 1/25$ being the same for both environments. However, DBAC is also able to solve `BipedalWalker-v3` with a single critic network with 5 times fewer critic parameters, but in an unstable way compared to TQC.

### 6.4 Ablation Study 4: Target Entropy Sensitivity

Target entropy of the policy is another hyper-parameter affecting performance. Therefore, different $\bar{\mathcal{H}}$ values are tested for three environments; `Hopper-v4`, `HalfCheetah-v4` and `Humanoid-v4`. The same $\beta$ values are used from the main experiment with dropout equal to 0.01. Learning curves are available in 9. With a smaller $\bar{\mathcal{H}}$ value, the agent in `HalfCheetah-v4` learns earlier but tends to demonstrate unstable behaviour after learning. According to the results, target entropy $\bar{\mathcal{H}}$ is not a critical parameter compared to $\beta$ for DBAC.

## 7 Prior Art

**Pessimistic policy evaluation**   Earlier approaches to overcome the overestimation bias phenomenon by using double critic networks (Van Hasselt et al., 2016; Wang et al., 2016), lagged critic networks (Lillicrap et al., 2015), and combination of them (Fujimoto et al., 2018; Haarnoja et al., 2018). Recent methods use an ensemble of critic networks are used to capture epistemic uncertainty, and pessimistic Bellman backup for critic training (Chen et al., 2021b; Lan et al., 2020; Kumar et al., 2019). Similarly, Hiraoka et al. (2021) utilized dropout for critic regularization on top of this approach to increase this capability. These methods use constant pessimism for policy evaluation and improvement but better estimated epistemic uncertainty allows them to use a high update-to-data (UTD) ratio, the number of gradient steps per environment interaction.

Unlike previous approaches, Kuznetsov et al. (2020) define pessimism for each environment separately, in the form of sample truncation from quantile distribution and critic ensemble together. While ensembling is for epistemic uncertainty, quantile representation captures aleatoric uncertainty, in which they stated: *it is especially useful for precise overestimation control*, supportive to our idea about it. Similarly, our method

uses a single critic network returning normal distribution instead of quantile representation and employs Bayesian dropout instead of ensemble.

Moskovitz et al. (2021) focus on updating pessimism *on the fly* as a bandit problem instead of fixing it but this requires evaluating on-policy returns and introduces an online bandit to update pessimism. This approach would work in an off-policy online setting surely, but it is not usable in a completely offline setting since there would be no feedback for the bandit. Li et al. (2023b) goes beyond this approach by parameterization of optimism/pessimism with a neural network and obtains significantly good performance on benchmarks. Still, these methods use critic ensembles, thus increasing computational overhead. Our work does not focus on pessimism adaptation and takes it as constant through training and focuses on well-estimated predictive critic uncertainty.

**Risk-sensitive reinforcement learning**   Pessimism is also needed for safety-critical RL applications to avoid catastrophic situations. For this purpose, pessimistic policy updates upon aleatoric uncertainty are modeled as normal distribution (Tang et al., 2019; Yang et al., 2021). Stachowicz & Levine (2024) devised a risk-sensitive actor-critic algorithm in which epistemic uncertainty is modeled by ensemble whereas aleatoric uncertainty is modeled by distributional representation as an output of critic network similar to the work of Kuznetsov et al. (2020). Their approach leads to higher performance by significantly reducing unsafe maneuvers.

**Overestimation vs overfitting**   Li et al. (2023a) demonstrated that statistical overfitting should be mitigated for efficient RL. This is a valid statement for any machine learning problem since overfitting means poor generalization. On the other hand, Kumar et al. (2019) and Levine et al. (2020) state that overestimation is different than statistical overfitting since increasing the number of training samples does not prevent it. However, temporary overfitting may cause bias, and it may not be recoverable in actor-critic settings. In our work, we adopt pessimism to overfitted state-action pairs, resolving overestimation similar to previous works (Chen et al., 2021b; Hiraoka et al., 2021; Kumar et al., 2019; Lan et al., 2020) but in a more computation efficient way.

**Optimism in the face of uncertainty**   Epistemic uncertainty is also employed to improve policy *optimistically* (Audibert et al., 2007; Kocsis & Szepesvári, 2006). However, for large-scale problems, this approach either fails or requires carefully tuned optimism (Pacchiano et al., 2020; Ciosek et al., 2019). O'Donoghue et al. (2018) used normal distribution to track critic uncertainty in which the upper bound is used as a policy improvement target. Osband et al. (2016) follows a similar way but uses ensembles, and improves policy with random critics at each episode inspired by Thompson sampling. However, their experiments are on relatively easier environments for deep RL, so overestimation correction is not a major bottleneck. In the actor-critic setting, Tasdighi et al. (2024a) implemented a double critic network and used optimistic estimates for policy learning while constructing pessimistic critic targets to mitigate the critic overestimation problem. Ciosek et al. (2019) followed a similar way by using optimistic estimates only for exploration and pessimistic critic targets using double critics.

**Dropout uncertainty**   Using dropout is a kind of Bayesian approximation, so another way to assess model uncertainty (Gal & Ghahramani, 2016). It has applications on model-based (Gal et al., 2016a; 2017; Kahn et al., 2017) and model-free (Moerland et al., 2017; Jaques et al., 2019; He et al., 2021) reinforcement learning. Using the same idea in off-policy maximum entropy actor-critic setting, He et al. (2021) injects dropout to the critic network and demonstrates that one critic network is enough for an actor-critic method. Similarly, Hiraoka et al. (2021) uses dropout mechanism to evaluate epistemic uncertainty additive to ensembling, and shows that it reduces the number of required networks in the ensemble but they used critics with deterministic head. Dropout allows us to significantly reduce ensemble size and use a high UTD ratio like Chen et al. (2021b). A high UTD ratio increases the risk of overestimation bias but policy is certainly improved in a more pessimistic way to handle it. They also experimented with a single critic network (called Sin-DroQ) and obtained similar performance in easier environments such as `Hopper-v2` and `Walker2d-v2` but failed to converge for `Ant-v2` and `Humanoid-v2`. We empower this approach by introducing a heteroscedastic critic network and bootstrapping dropout uncertainty into a distributional representation.

**Heteroscedastic representation for epistemic uncertainty**   Although heteroscedastic representation is mainly used to assess aleatoric uncertainty, it may also capture epistemic uncertainty if trained for this purpose, such as evidential learning (Sensoy et al., 2018; Amini et al., 2020). Moreover, Lakshminarayanan et al. (2017) also used ensembles to create adversarial examples to robustly assess predictive uncertainty as neural network output. We treat critic estimate as a random value both epistemically (sourced by dropout of TD target) and aleatorically (sourced by state transition and learning dynamics) and use distributional representation to learn both.

## 8   Conclusion & Future Directions

In this paper, we introduced Deep Bayesian Actor-Critic (DBAC), a novel off-policy actor-critic algorithm. The main idea is to inhibit overestimation for the sake of faster and more robust learning by incorporating critic uncertainty arising from both limited samples (epistemic uncertainty) and environmental stochasticity (aleatoric uncertainty). We utilize Bayesian dropout to capture epistemic uncertainty, while heteroscedastic output models the total predictive uncertainty which serves to pessimistic critic and policy updates, enabling stability and robustness in learning. Moreover, we used normal distribution to represent predictive critic uncertainty, but our analysis is valid for all sub-Gaussian critic distributions or quantile representations. Finally, we derived an upper bound for overestimation, demonstrating that an adequate level of pessimism mitigates overestimation without succumbing to underestimation, thus facilitating computation and sample-efficient learning.

**Limitations & Future work**   Our ablation studies demonstrate the effects of dropout rate and pessimism, revealing the sensitivity of the learning procedure to these parameters. For each specific environment and optimization method, there exists an optimal level of pessimism and dropout rate. A promising direction for future research is to develop a grounded method to adjust the pessimism level for specific environments and agents to allow better adaptation for the learner to the environment. In addition, the sensitivity of similar algorithms to pessimism and dropout rate should be investigated in depth. Another promising direction for future work is to explore different methods for tracking epistemic uncertainty other than ensembles and Bayesian dropout. Bayesian neural networks (Depeweg et al., 2016), concrete dropout (Gal et al., 2017), and evidential deep learning (Sensoy et al., 2018; Amini et al., 2020) frameworks may offer more computation efficient alternatives. Lastly, uncertainty calibration of critic estimates is worth investigating, which might make pessimism tuning much easier.

**Broader Impact**   DBAC tackles critical challenges such as accelerating learning, improving stability, and ensuring computation efficiency. Our research not only pushes the boundaries of reinforcement learning but also promises significant implications for enhancing the safety and intelligence of robots, self-driving cars, and autonomous systems in healthcare and finance.

**Acknowledgments**

This research has received no external funding.

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

# Appendix A   Proofs

*Proof of Theorem 4.1.* Analyzing Bellman update $\mathcal{T}^*\mathcal{Q}(s,a)$,

$$
\begin{aligned}
\mathcal{T}^*\mathcal{Q}(s,a) &= R(s,a) + \gamma \mathbb{E}_{s'\sim\tau(\cdot|s,a)}\Big[\mathbb{E}_{q\sim\mathcal{Q}}\Big[\alpha\log\Big(\int_{\mathcal{A}}\exp(\alpha^{-1}q(s',a'))da'\Big)\Big]\Big] \\
&\leq R(s,a) + \gamma \mathbb{E}_{s'\sim\tau(\cdot|s,a)}\Big[\alpha\log\Big(\int_{\Omega_\mathcal{Q}}\int_{\mathcal{A}}\exp(\alpha^{-1}q(s',a'))da'd\mathcal{Q}\Big)\Big] \\
&= R(s,a) + \gamma \mathbb{E}_{s'\sim\tau(\cdot|s,a)}\Big[\alpha\log\Big(\int_{\mathcal{A}}\int_{\Omega_\mathcal{Q}}\exp(\alpha^{-1}q(s',a'))d\mathcal{Q}da'\Big)\Big] \\
&\leq R(s,a) + \gamma \mathbb{E}_{s'\sim\tau(\cdot|s,a)}\Big[\alpha\log\Big(\int_{\mathcal{A}}\exp(\alpha^{-1}\mu(s',a') + \tfrac{1}{2}\alpha^{-2}\sigma^2(s',a'))da'\Big)\Big] \\
&\leq R(s,a) + \gamma \mathbb{E}_{s'\sim\tau(\cdot|s,a)}\Big[\alpha\log\Big(\big(\int_{\mathcal{A}}\exp(\alpha^{-1}\mu(s',a'))da'\big)\cdot\big(\max_{a'}\exp(\tfrac{1}{2}\alpha^{-2}\sigma^2(s',a'))\big)\Big)\Big] \\
&= R(s,a) + \gamma \mathbb{E}_{s'\sim\tau(\cdot|s,a)}\Big[\alpha\log\Big(\int_{\mathcal{A}}\exp(\alpha^{-1}\mu(s',a'))da'\Big) + \frac{1}{2\alpha}\max_{a'}\sigma^2(s',a')\Big] \\
&= R(s,a) + \gamma \mathbb{E}_{s'\sim\tau(\cdot|s,a)}\Big[\alpha\log\Big(\int_{\mathcal{A}}\exp(\alpha^{-1}\mu(s',a'))da'\Big)\Big] + \frac{\gamma}{2\alpha}\mathbb{E}_{s'\sim\tau(\cdot|s,a)}\Big[\max_{a'}\sigma^2(s',a')\Big].
\end{aligned}
$$

First inequality comes from Jensen's inequality (using concave property of log function) while the following equality is a result of Tonelli's theorem. The second inequality results from the property of sub-Gaussian distribution 4.1, where the first statement of the theorem is proven. The following inequality is a result of the mean value theorem for integrals. In the last equality, the first two terms are equal to $\mathcal{T}^*\mu(s,a)$. Therefore,

$$
\epsilon(s,a) = \mathcal{T}^*\mathcal{Q}(s,a) - \mathcal{T}^*\mu(s,a) \leq \frac{\gamma}{2\alpha}\mathbb{E}_{s'\sim\tau(\cdot|s,a)}\Big[\max_{a'}\sigma^2(s',a')\Big].
$$

$\square$

*Proof of Corollary 4.1.1.* From the Theorem 4.1, we can show that

$$
\begin{aligned}
\mathcal{T}^*\tilde{\mathcal{Q}}(s,a) &\leq R(s,a) + \gamma \mathbb{E}_{s'\sim\tau(\cdot|s,a)}\Big[\alpha\log\Big(\int_{\mathcal{A}}\exp(\alpha^{-1}(\mu(s',a') - \beta\sigma(s',a') + \tfrac{1}{2}\alpha^{-1}\sigma^2(s',a')))da'\Big)\Big] \\
&= R(s,a) + \gamma \mathbb{E}_{s'\sim\tau(\cdot|s,a)}\Big[\alpha\log\Big(\int_{\mathcal{A}}\exp(\alpha^{-1}\mu^\dagger(s',a'))da'\Big)\Big] = \mathcal{T}^*\mu^\dagger(s,a).
\end{aligned}
$$

where we have defined $\mu^\dagger(s',a') = \mu(s',a') - \beta\sigma(s',a') + \tfrac{1}{2}\alpha^{-1}\sigma^2(s',a'))$. If $\beta \geq \max_{(s',a')}\tfrac{1}{2}\alpha^{-1}\sigma(s',a')$, then $\mu^\dagger(s',a') < \mu(s',a')$. So we can show that

$$
\mathcal{T}^*\tilde{\mathcal{Q}}(s,a) \leq \mathcal{T}^*\mu^\dagger(s,a) \leq \mathcal{T}^*\mu(s,a). \tag{23}
$$

$\square$

# Appendix B    Results of Ablation Studies

## B.1    Ablation Study 1: Pessimism Sensitivity

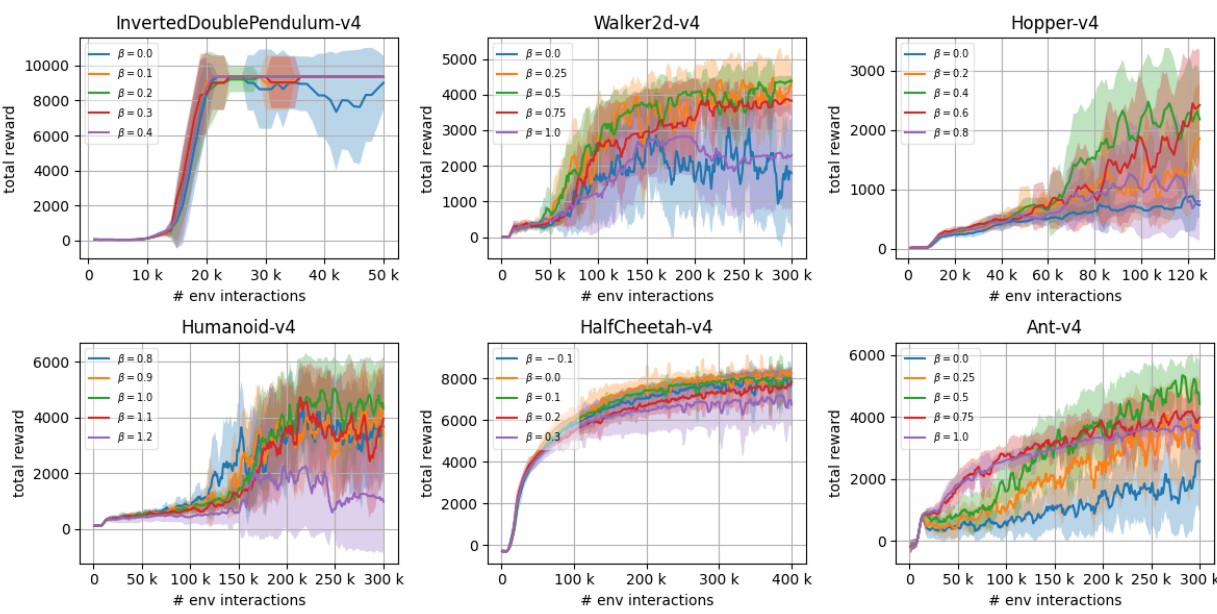

Figure 3: Learning curves of DBAC with varying pessimism ($\beta$) parameter. Dropout is equal to 0.01 for all experiments.

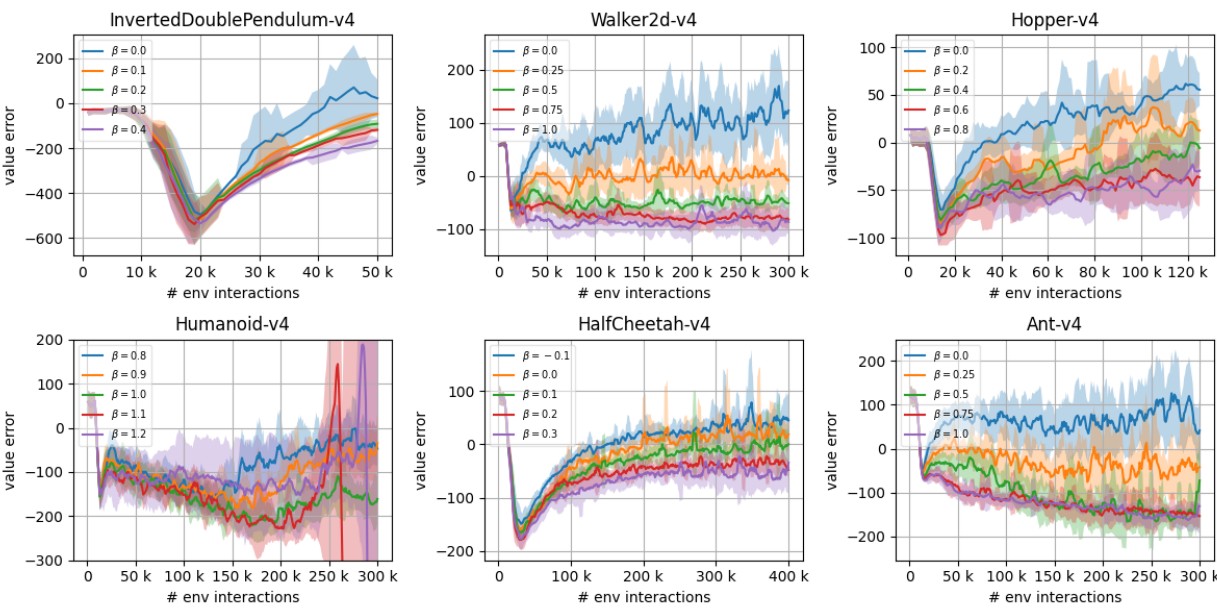

Figure 4: Episodic value estimation error curves of DBAC with varying pessimism ($\beta$) parameter. Dropout is equal to 0.01 for all experiments.

## B.2 Ablation Study 2: Dropout Sensitivity

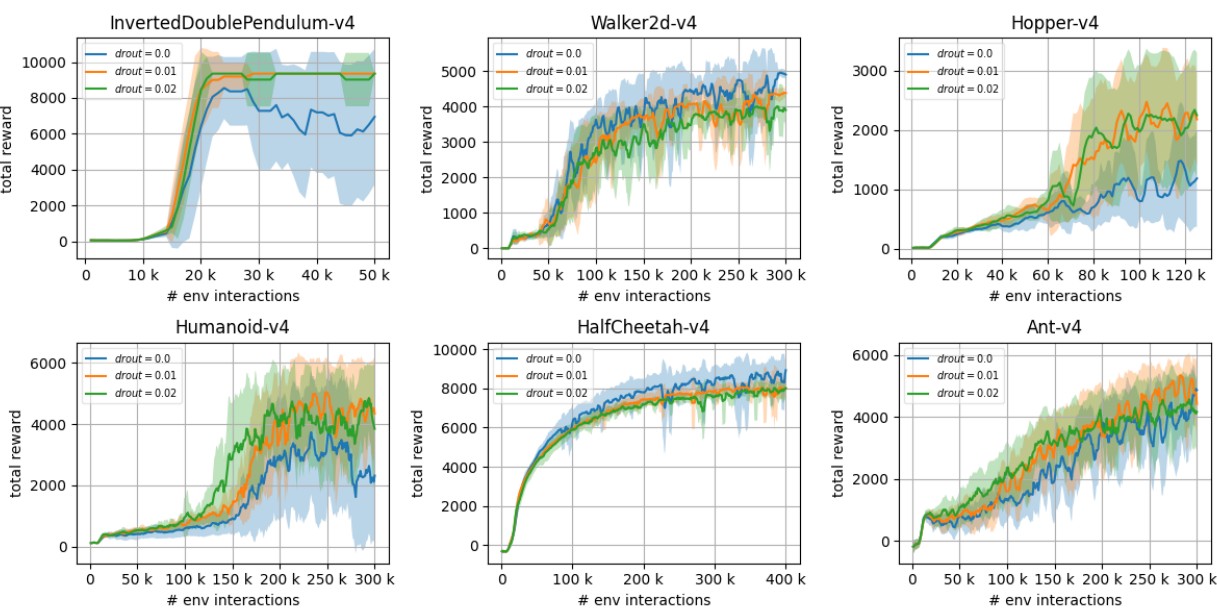

Figure 5: Learning curves of DBAC with varying dropout rates (for both critic and policy). Pessimism parameters are used the same as the main experiment, available in Table 5.

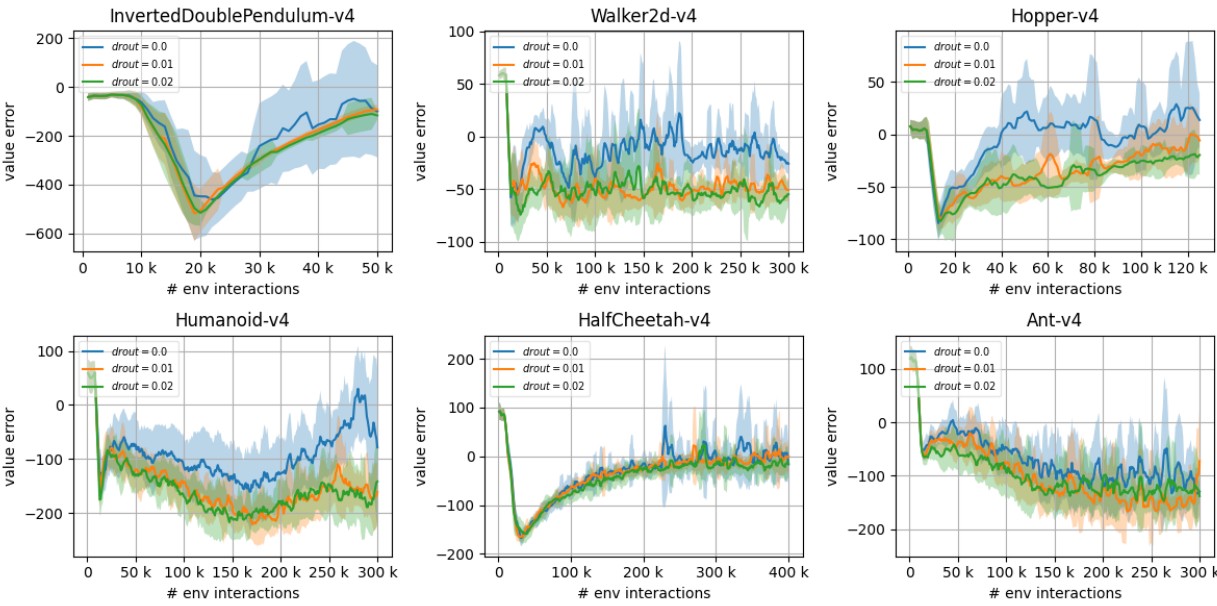

Figure 6: Episodic value estimation error curves of DBAC with varying dropout rates (for both critic and policy). Pessimism parameters are used the same as the main experiment, available in Table 5.

### B.3 Ablation Study 3: Experiments on Stochastic Environments

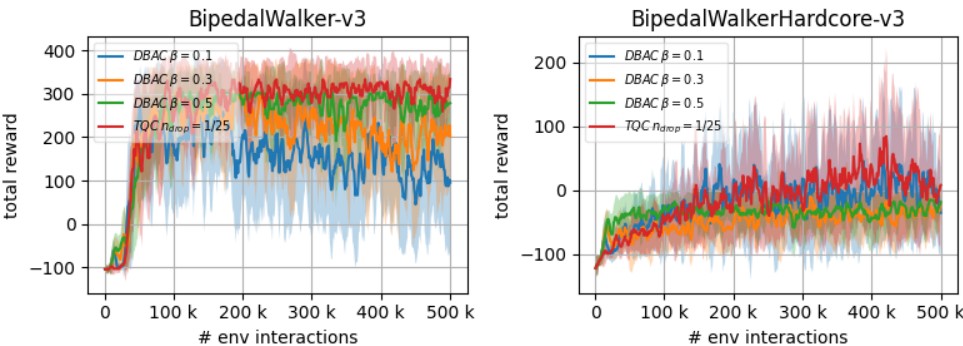

Figure 7: Learning curves of DBAC with varying $\beta$ and TQC ($n_{drop} = 1/25$), on Bipedal Walker environments, $\bar{H} = -2$.

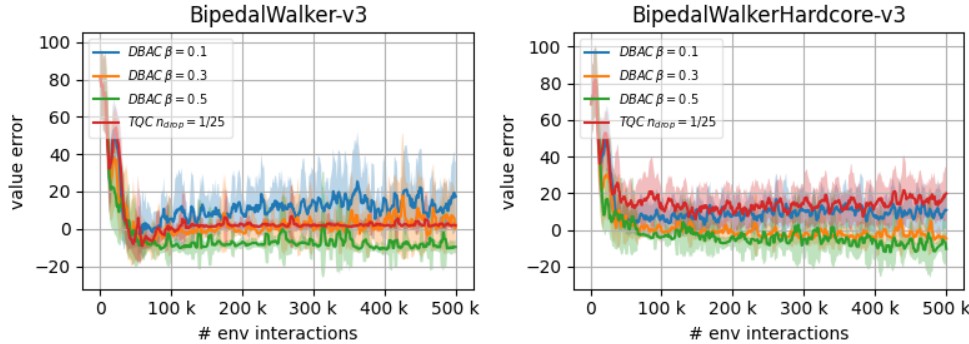

Figure 8: Episodic value estimation error curves of DBAC with varying $\beta$ and TQC ($n_{drop} = 1/25$), on Bipedal Walker environments, $\bar{H} = -2$.

### B.4 Ablation Study 4: Target Entropy Sensitivity

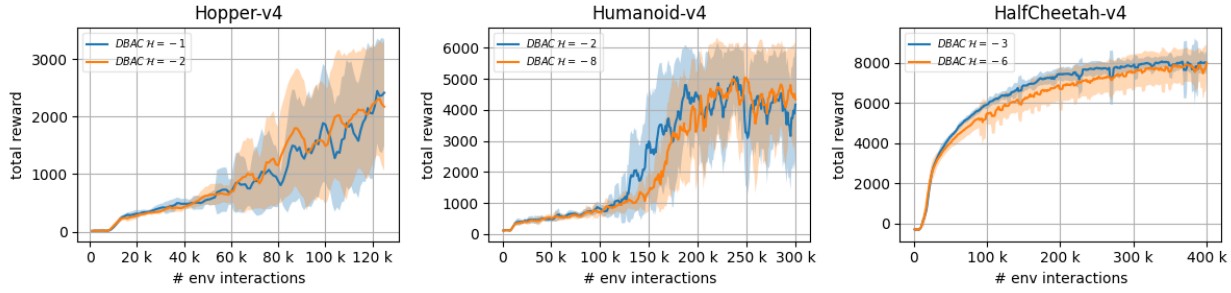

Figure 9: Learning curves of of DBAC with different target policy entropy $\bar{H}$. Dropout is 0.01 and pessimism parameters are used same to main experiment, available in Table 5.

# Appendix C   Hyper-parameters and Experiment Details

Hyper-parameter values used in the experiments per method are listed in Table 4. Dropout parameter is found by trial-and-error and it matches the selection in DROQ paper (Hiraoka et al., 2021). In addition, target entropy and pessimism parameters (only for DBAC) are summarized in Table 5. Target entropy values are taken from the DROQ paper, which uses the same values (except `Humanoid-v4`). For DBAC, pessimism hyper-parameter and for TQC, quantile drop parameters per environment are found by trial-and-error to obtain the best performance.

Table 4: Experimental Parameters per Algorithm

| Algorithm | Parameter | Value |
|---|---|---|
| DBAC, DROQ, SAC, TOPSAC, TQC | Optimizer | Adam ((Kingma & Ba, 2014)) |
| | Critic Learning Rate | $1 \times 10^{-3}$ |
| | Actor Learning Rate | $3 \times 10^{-4}$ |
| | Discount Rate ($\gamma$) | 0.99 |
| | Target-Smoothing Coefficient ($\rho$) | 0.995 |
| | Replay Buffer Size | $1 \times 10^{6}$ |
| | Mini-Batch Size | 256 |
| | Random Starting Data | 10000 |
| | UTD Ratio ($G$) | 1 |
| DROQ | Dropout Rate | 0.01 |
| TOPSAC, TQC | Number of Quantiles | 25 |
| TQC | Ensemble Size | 5 |
| TOPSAC | Bandit Optimism/Pessimism Arms | [-1, -0.5, 0] |
| | Bandit Learning Rate | 0.1 |
| | Bandit Window Size | 10 |

Table 5: Target policy entropy ($\bar{H}$), pessimism ($\beta$ for DBAC), dropout rate (for DBAC) and quantile drop ($n_{drop}$ for TQC) per Environment

| Environment | Entropy ($\bar{H}$) | Pessimism ($\beta$) | Dropout | Quantile Drop ($n_{drop}$) |
|---|---|---|---|---|
| Ant-v4 | -4 | 0.5 | 0.01 | 5/25 |
| Hopper-v4 | -1 | 0.6 | 0.01 | 5/25 |
| Walker2d-v4 | -3 | 0.5 | 0.00 | 5/25 |
| HalfCheetah-v4 | -3 | 0.1 | 0.00 | 0/25 |
| Humanoid-v4 | -8 | 1.0 | 0.01 | 12/25 |
| InvertedDoublePendulum-v4 | -1 | 0.2 | 0.01 | 3/25 |

## Appendix D    Network Architectures of DBAC

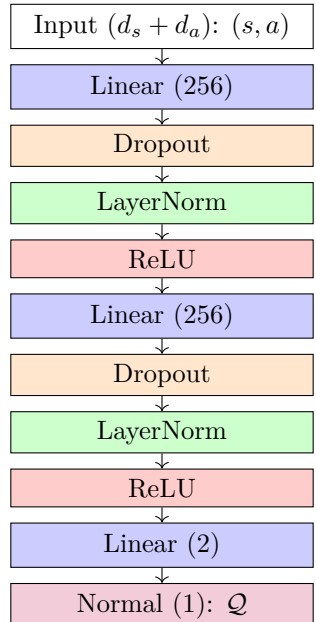

Figure 10: Critic network architecture

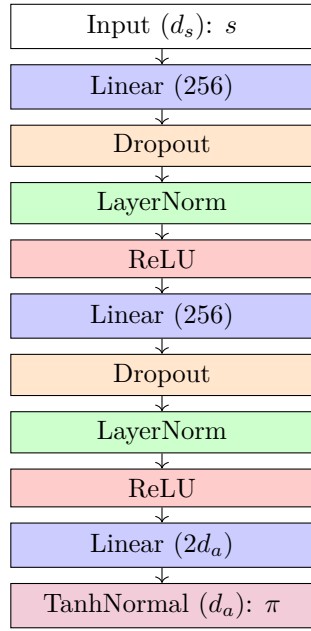

Figure 11: Policy network architecture

## Appendix E    Source Code

Our results can be accessed publicly at `https://github.com/authors-github/deep-bayesian-actor-critic-results`. This code uses our in-house developed RL framework as a sub-repository, available on `https://github.com/authors-github/rl-warehouse`.

