# OpenReview forum: "Computation and Sample Efficient Reinforcement Learning with Deep Bayesian Actor-Critic"
_TMLR — Rejected by TMLR_

### Review · Reviewer_22Vj · 2024-09-02

**Summary Of Contributions:**

The authors propose and evaluate a Bayesian Deep RL approach which combines a distributional Q function approximation with bayesian dropout to model both aleatoric and epistemic uncertainty of the value function.

**Audience:**

Yes

**Claims And Evidence:**

No

**Requested Changes:**

Cite Moskovitz et al in Section 4 and compare the insights.

Provide a wider setup for the ablations so that the impact of the parameter $\beta$ can properly be understood.

Provide experiments for the calibration of the uncertainty estimation compared to an ensemble method.

# Questions

Why do the authors only consider a Gaussian parameterization for the Q value estimate instead of a more complex form like the quantiles chosen by Moskovitz?

**Strengths And Weaknesses:**

The paper outlines the usefulness of modelling both epistemic and aleatoric uncertainty in the context of deep RL.

The paper is mostly very readable without unnecessary complications, but does contain some instances of incorrect grammar which should be cleaned up, e.g. Beginning of 5.5: Finally, we present THE algorithm [...]

# Weaknesses and concerns
The main weakness is the limited impact of the theoretical section on the experiments as well as the limited experimental section.

The main theoretical contribution is highly similar to that of Moskovitz et al (cited in the paper). The idea of pessimism through variance estimation is both theoretically and in terms of implementation strikingly similar. This similarity is not acknowledged in section 4, which is a problematic oversight.

Furthermore, while the parameter $\beta$ is important for guaranteeing limited overestimation, in the implementation this parameter is hand tuned per environment. While the appendix includes some ablations over this parameter, these cover different ranges for different environments, meaning no overall idea of the sensitivity of the algorithm on the parameter can be gleaned.

The dropout parameter is set to 0 in some environments, which leads to the method not actually having any meaningful interpretation as a Bayesian approach if I understand the paper correctly?

This also complicates the comparison with other works, as tuning the $\beta$ and dropout parameters likely involves additional samples (in the form of a hyperparameter sweep) which are unreported here. The authors specifically state that the bandit approach chosen by Moskovitz is inefficient due to requiring additional samples for the bandit evaluation, but this is not an even comparison.

Furthermore, given the claim of the paper that both aleatoric and epistemic uncertainty are important to capture, I would encourage the authors to evaluate on stochastic environments. The OpenAI Gym locomotion tasks are mostly deterministic with the only stochasticity coming from numerical issues in the simulator.

In terms of strengthening the paper, I would encourage the authors to focus more on explanatory experiments. For example, it would be very useful to have an experiment that focuses on the question whether the uncertainty estimation is well calibrated, high when the error is low. The authors plot the average value error, which is useful, but does not paint the full picture.

Due to the similarity between the approach presented here and the one presented by Moskovitz et al, I would encourage the authors to add the respective reward curves for ease of comparison.

---

> ### Author Response · Authors · 2024-09-23
>
> Thanks for the valuable comments. We tried to answer your questions and adjust our work according to your recommendations.
>
> **Q1)** The main theoretical contribution is highly similar to that of Moskovitz et al (cited in the paper). The idea of pessimism through variance estimation is both theoretically and in terms of implementation strikingly similar. This similarity is not acknowledged in section 4, which is a problematic oversight.
>
> **A1)** Moskovitz et al. (TOP) is properly cited at the end of section 4.
>
> **Q2)** Furthermore, while the parameter is important for guaranteeing limited overestimation, in the implementation this parameter is hand-tuned per environment. While the appendix includes some ablations over this parameter, these cover different ranges for different environments, meaning no overall idea of the sensitivity of the algorithm on the parameter can be gleaned.
>
> **A2)** Due to a lack of computational resources, we found the best pessimism hyper-parameter (beta) by hand-tuning and sweeped in the ablation study to show that optimal pessimism is neither a higher or lower number. We mentioned it on future work part.
>
> **Q3)** The dropout parameter is set to 0 in some environments, which leads to the method not actually having any meaningful interpretation as a Bayesian approach if I understand the paper correctly?
>
> **A3)** In the dropout ablation, some environments work best with zero dropout. We commented about it as they do not suffer from critic overestimation a lot.
>
> **Q4)** This also complicates the comparison with other works, as tuning the and dropout parameters likely involves additional samples (in the form of a hyperparameter sweep) which are unreported here. The authors specifically state that the bandit approach chosen by Moskovitz is inefficient due to requiring additional samples for the bandit evaluation, but this is not an even comparison.
>
> **A4)** Tuning hyper-parameters surely requires parameter sweeps. We do not claim the bandit approach of TOP algorithm is inefficient but state that it requires to evaluation of on-policy cumulative returns and cannot be deployed to work completely offline setting since there is no feedback of returns for the bandit. We adjusted our comment about the usage of bandit in TOP for pessimism tuning, in the Prior Art section.
>
> **Q5)** Furthermore, given the claim of the paper that both aleatoric and epistemic uncertainty are important to capture, I would encourage the authors to evaluate on stochastic environments. The OpenAI Gym locomotion tasks are mostly deterministic with the only stochasticity coming from numerical issues in the simulator.
>
> **A5)** We argue that non-stationary nature of the learning procedure creates stochasticity of value estimates and appears as aleatoric uncertainty, as we explained in section 5. Still, we created a new ablation study to compare DBAC and TQC in stochastic environments. We used BipedalWalker-v3 and BipedalWalkerHardcore-v3 environments since these are both stochastic (randomly generated terrain).
>
> **Q6)** In terms of strengthening the paper, I would encourage the authors to focus more on explanatory experiments. For example, it would be very useful to have an experiment that focuses on the question whether the uncertainty estimation is well calibrated, high when the error is low. The authors plot the average value error, which is useful, but does not paint the full picture.
>
> **A6)** First, we do not claim that critic estimates are well calibrated but captures epistemic and aleatoric uncertainty up to some level. Even we would try so, as the learning is in regression form and learning procedure is not stationary, we do not know how to assess uncertainty calibration of critic. We commented about it on the future work and 5.2.
>
> **Q7)** Due to the similarity between the approach presented here and the one presented by Moskovitz et al, I would encourage the authors to add the respective reward curves for ease of comparison.
>
> **A7)** We added results of TOP algorithm as SAC variant (for fair comprasion).
>
> **Q8)** Why do the authors only consider a Gaussian parameterization for the Q value estimate instead of a more complex form like the quantiles chosen by Moskovitz?
>
> **A8)** For simplicity. As we found the overestimation in standard deviation form, we conducted experiments by modeling the critic estimate as a normal distribution. However, it is still possible to use other distributions and quantile representations.
>
> Requested Changes:
>
> - Cite Moskovitz et al in Section 4 and compare the insights. **Done**
>
> - Provide a wider setup for the ablations.
>
> In ablation studies, value error curves are added and discussed. We will extend pessimism ablation with five different values. We aim to finish this before the deadline.
>
> - Provide experiments for the calibration of the uncertainty estimation compared to an ensemble method.
>
> We could not do this due to theoretical complexity, explained in answer 6.

---

### Review · Reviewer_jMYq · 2024-09-04

**Summary Of Contributions:**

The paper introduces Deep Bayesian Actor-Critic (DBAC), which is designed to address the challenges of sample and computational inefficiency in actor-critic methods. Traditional actor-critic methods, while efficient in sample usage, often suffer from overestimation bias, which is typically countered by implementing pessimistic policy evaluation. However, this approach can lead to underestimation bias and requires careful tuning, especially when using ensemble methods to represent critic uncertainty, which also increases computational demands.

To overcome these limitations, the authors replace the ensemble approach with a single critic network that incorporates Bayesian dropout and a heteroscedastic network. This design allows the agent to become uncertainty-aware and mitigates overestimation bias, and also the use of dropout and a distributional representation of the critic enhances computational efficiency. As a result, the method achieves high sample and computation efficiency. with performance near the state-of-the-art levels.

**Audience:**

Yes

**Claims And Evidence:**

Yes

**Requested Changes:**

- Consider incorporating a more adaptive mechanism for tuning these parameters dynamically during training. Or please have some discussion related to this as a future work

**Strengths And Weaknesses:**

***Strength***
- Computational Efficiency: the proposed method significantly reduces the computational overhead compared to traditional ensemble-based methods by using a single critic network with Bayesian dropout.
- Easy way of adapting pessimism: the proposed method incorporates an adjustable pessimism parameter, which helps in mitigating overestimation bias.
- Robustness and Stability: The combination of bayesian dropout and a distributional representation of the critic's output contributes to the stability of the learning process, make the algorithm more stable to the challenges posed by noisy data and non-stationary environments.
- Simplicity: Overall the algorithm is relatively easy to implement given the current actor critic algorithms.

***Weakness***
- Sensitivity: The performance of the algorithm is highly sensitive to the tuning of key hyperparameters, particularly the pessimism and dropout rates.
- More robust / automatic way of tunning hyperparametrs: if we can find a more automated or adaptive approach to parameter selection, the algorithm could be easily verified on new / unseen tasks.

---

> ### Author Response · Authors · 2024-09-23
>
> Glad to receive this valuable feedback. After this feedback,
>
> - In the new version, we highlighted the drawbacks of the DBAC algorithm which are sensitivity to pessimism and dropout in detail in future directions and discussion part.
>
> - Pessimism hyper-parameter tuning is also discussed in detail in the future direction part.

---

### Review · Reviewer_97c1 · 2024-09-18

**Summary Of Contributions:**

This paper considers actor-critic algorithms in reinforcement learning. It explains that the existence of uncertainty in the critic may result in overestimation, and therefore a pessimistic evaluation of policy should be used. The most common way is to use ensembles but that is computationally expensive. So this paper proposes using Bayesian dropouts and a heteroscedastic critic network instead.

**Audience:**

No

**Claims And Evidence:**

No

**Requested Changes:**

The papers should clearly state the setting and the problem it is trying to solve, as well as the contributions. The mix of pessimism and optimism discussion is particularly confusing, as the former is typically used in offline RL and the latter is used in online RL.

**Strengths And Weaknesses:**

Strengths:
- Uncertainty estimation using dropout and a single critic network is indeed more computationally efficient than ensembles.

Weaknesses:
- I find the writing convoluted and confusing, making it difficult to understand the problem this paper is trying to solve. Just the first two paragraphs of Introduction include a mix of challenges in RL: sample efficiency, computational efficiency, generalization, deadly triad, inductive bias (what type of inductive bias?), off-policy updates, overestimation, underestimation, etc. None of these terms are properly defined and the setting and motivation are unclear.
- What is the optimism-pessimism dilemma? This is not a recognized dilemma in RL to my knowledge, and the section does not provide a clear explanation. It seems that the paper is considering online RL with off-policy updates, and perhaps the goal is to use optimism to conduct exploration and pessimism to prevent overestimation in using off-policy samples. Overestimation in the online setting is not inherently bad, this is precisely why an exploration bonus is added in equations (2) and (3). The paper does not clarify what type of overestimation might be harmful.
- The problem and setting are not at all formulated clearly. The paper should clearly state what are the knowns and unknowns of the problem and data collection procedure.
- I don't find this sentence to be correct in the online setting: "However, when most out-of-distribution estimates are not consistent with reality (poor generalization), they need to be pessimistic and should not trust their estimate unless sufficient observations are obtained." In the online setting, optimism is appropriate as it encourages getting more observations in the unknown regions.
- Novelty and contributions seem limited to using uncertainty estimation via dropout instead of ensembles. Empirical results are not strong.

---

> ### Author Response · Authors · 2024-09-23
>
> Thanks for this valuable feedback. Accordingly, we tried to explain our idea more clearly and tried to answer the comments and questions as much as possible.
>
> **Q1)** I find the writing convoluted and confusing, making it difficult to understand the problem this paper is trying to solve. Just the first two paragraphs of Introduction include a mix of challenges in RL: sample efficiency, computational efficiency, generalization, deadly triad, inductive bias (what type of inductive bias?), off-policy updates, overestimation, underestimation, etc. None of these terms are properly defined and the setting and motivation are unclear.
>
> **A1)** In the new version, we tried to explain that our learning setting is online off-policy RL with temporal difference and defined the terms again as clear as possible. By inductive bias, we meant the generalization capability and changed the text accordingly.
>
> **Q2)** What is the optimism-pessimism dilemma? This is not a recognized dilemma in RL to my knowledge, and the section does not provide a clear explanation. It seems that the paper is considering online RL with off-policy updates, and perhaps the goal is to use optimism to conduct exploration and pessimism to prevent overestimation in using off-policy samples. Overestimation in the online setting is not inherently bad, this is precisely why an exploration bonus is added in equations (2) and (3). The paper does not clarify what type of overestimation might be harmful.
>
> **A2)** We refined our text, stating that optimism is necessary for exploration of state space but pessimism is necessary to prevent critic value overestimation, in off-policy temporal difference learning. We tried to define optimism-pessimism dilemma as the agent should balance them, similar to exploration-exploitation dilemma. However, as you stated, there is no such recognized dilemma and changed the title to `Pessimism for Actor-Critic`.
>
> **Q3)** The problem and setting are not at all formulated clearly. The paper should clearly state what are the knowns and unknowns of the problem and data collection procedure.
>
> **A3)** In the new version, we explained how online off-policy actor-critic works with components in detail, such as actor, critic, experience buffer etc., in section 1.2. In the section 1.2, we clearly stated contributions.
>
> **Q4)** I don't find this sentence to be correct in the online setting: "However, when most out-of-distribution estimates are not consistent with reality (poor generalization), they need to be pessimistic and should not trust their estimate unless sufficient observations are obtained." In the online setting, optimism is appropriate as it encourages getting more observations in the unknown regions.
>
> **A4)** In online setting, optimistic learning works better surely, as it encourages exploration of state-action space. However, even in online setting, off-policy learning with temporal difference yields critic overestimation which is not recoverable. This is also explained in mentioned algoritms (SAC, DROQ, TQC, TOP etc.) which are in same setting, i.e, online off-policy actor critic. We tried to explain it on subsection 1.1.
>
> **Q5)** Novelty and contributions seem limited to using uncertainty estimation via dropout instead of ensembles. Empirical results are not strong.
>
> **A5)** In addition to dropout, we argue that aleatoric uncertainty is also important and overall predictive uncertainty of critic is assesed through distributional representation. Moreover, we do not claim to beat other algorithms in performance but offer a more computation and sample efficient online off-policy actor-critic algorithm.

---

### Author Response · Authors · 2024-09-23
**Changes in new version**

We again thank to all reviewers for their feedbacks and suggestions. Other than typo and grammar corrections, changes are as follows;

- The TOP paper (Moskovitz et al.) is cited in the 4th section.
- A more detailed explanation of why zero dropout is used for HalfCheetah-v4 and Walker2d-v4.
- Some additions to future work about pessimism - dropout sensitivity, uncertainty calibration etc.
- We implemented the TOP algorithm with SAC variant (to make fair comprasion to other maximum entropy methods), named it as TOPSAC run for 6 MuJoCo environments and added respective reward curves.
- Value error curves for ablation studies are added to get more insights about pessimism and dropout.
- We explained why aleatoric uncertainty modeling is necessary even in determinstic environments.
- Third ablation study is conducted for two stochastic environments, by comparing DBAC and TQC.
- As we previously conducted target entropy ablation, we added them as 4th ablation study.
- Network architectures are visualized in Appendix.

In short time, we will be extending pessimism parameter ablation, using 5 beta parameter instead of 3.

---

> ### Author Response · Authors · 2024-10-06
> **Minor updates**
>
> The revision is updated two times. To summarize, here are the updates;
>
> - Pessimism ablation is extended by using 5 parameters for each environment.
> - Minor grammar improvements and fixes.
> - Realized that Theorem 4.1 (inside the text) has missing reward term, this is fixed.
> - In section 1.2, learning setting it is stated that DBAC employs experience buffer (again, forgotten to add before).

---

### Decision · Action_Editor_Qz6w · 2024-11-04

**Recommendation:** Reject

**Comment:**

Most of the reviewers agreed that the idea of dropout instead of ensembles was simpler and could be less computationally intensive.

However, two of the three reviewers felt that their concerns were not addressed adequately following the responses.

The paper is difficult to read in parts using some confusing language with unexplained terms. The presentation can be improved.

The approach is too similar to an existing paper and the empirical results were too limited, hence the reviewers remain unconvinced that the paper is worthy of publication at this time. Please take note that it is not the volume of the empirical results that is the problem but rather the lack of evidence that using dropout over ensembles is worthwhile was not sufficiently demonstrated.

The reviewers have given a number of other helpful points of feedback.  I encourage the authors to develop the paper further based on these points, pursue more convincing set of experimental results, and resubmit an improved paper.

**Audience:**

This work is appropriate for the TMLR audience.

**Claims And Evidence:**

Two of the reviewers did not feel that the claims were supported by the evidence.

The contributions were limited to applying dropout rather than ensembles, and the method overall resembled existing work (Moskovitz et al.) very closely. The paper argued that two forms of uncertainty (epistemic and aleatoric) are important and then tested on mostly deterministic domains. The method was framed as a Bayesian method, but then in some cases the dropout parameter was set to 0, undermining the motivation.

Overall, the empirical results were not strong and seemed sensitive to hyper-parameters. It is true that the dropout approach is less computationally intensive than ensembles, but then if you have to sweep over the dropout hyper-parameter in every domain to confirm these gain before trusting it, it is not clearly an overall computational savings.